# Characterization of the murine spine for spaceflight studies

Shiyin Lim[1], Joanna E. Veres[2], Eduardo A.C. Almeida[3], Grace D. O'Connell[1,2]*

1 Department of Mechanical Engineering, University of California, Berkeley, California, United States of America, 2 Department of Bioengineering, University of California, Berkeley, California, United States of America, 3 NASA Ames Research Center, Moffett Field, California, United States of America

* g.oconnell@berkeley.edu

## Abstract

Rodents provide a useful analog for understanding the effects of spaceflight on the human body, offering opportunities for investigations into the relationship between microgravity and the musculoskeletal system. In particular, rodents have often been utilized to improve our understanding of the effects of spaceflight on the spine, including intervertebral disc and vertebral body health. However, there are a number of experimental factors that differ between existing works, including mission duration, animal housing, and anatomical location of interest, making it difficult to draw holistic conclusions. Additionally, the quadrupedal nature of the murine spine results in different biomechanical loading than in a bipedal organism. Thus, the objective of this study was to more fully define the bulk properties of the murine lumbar spine model after 28 days of spaceflight. Additionally, the proximal tibia was analyzed to provide insight into the skeletal site-specificity of gravitational unloading in space. Results indicated that the effects of spaceflight on vertebral body bone microarchitecture, intervertebral disc biochemistry, and intervertebral disc joint mechanics were statistically insignificant, while large and significant bone loss was observed in the proximal tibia of the same animals. We hypothesize that this may be due to site-specific loading changes in space. Specifically, vigorous ambulatory behaviors observed in this experiment after initial acclimation to spaceflight may increase axial load-bearing in the lumbar spine, while maintaining microgravity induced mechanical unloading in the tibia. In total, this work shows that the rodent spine, unlike the weight bearing tibia in the same mice, is not affected by gravitational unloading, suggesting the tissue degenerative effects of spaceflight are site- and load-specific and not systemic. This study also highlights the importance of considering experimental variables such as habitat acclimation, physical activity, and experiment duration as key factors in determining musculoskeletal and spine health outcomes during spaceflight.

**Data availability statement:** All relevant data are within the paper and its Supporting Information files.

**Funding:** This study was funded by the National Science Foundation (1751212, GDO) and the National Aeronautics and Space Administration (NNH14ZTT001N 14-14SF Step2-0063, EACA). The funders played no role in the study design, data collection, analysis and interpretation of data, decision to publish, or preparation of this manuscript.

**Competing interests:** The authors declare no competing interests.

## Introduction

As NASA prepares to send astronauts into lunar orbit and deep space, the biological implications of long duration space travel are becoming increasingly relevant. Previous studies in Low Earth Orbit (LEO), have indicated that spaceflight leads to reductions in bone mineral density (BMD) of the lumbar vertebrae, muscle atrophy in the trunk, and increases in disc herniation risk [1–3]. While these studies provided valuable data for improving our understanding of the relationship between spaceflight and human spine health, they are also limited by the types of analyses that can be performed and very small sample sizes. Therefore, rodents have long-served as an analog for studying the impact of long-duration spaceflight on tissue health.

In rodents, spaceflight has been shown to affect both the vertebral body and intervertebral disc, with reductions in bone volume fraction [4–6], alterations in disc biochemical composition [7,8], and changes in disc joint mechanics [9–11]. However, the experimental conditions across these studies vary widely. For example, flight duration has evolved from days to weeks as access to space improves, with the most recent International Space Station (ISS) rodent studies ranging from one-month [6,7,11–13], to two months (RR-5, RR-6), and 76 days on RR-7. Furthermore, some studies focus on the caudal spine, while other studies focus on the lumbar spine, making direct comparison challenging [5,6,9–11]. Another factor that is highly variable is rodent age, where animals may be anywhere from days to months old, and age has been shown to affect rodent responses to spaceflight [7,12,13]. Furthermore, rodent age may affect meaningful translation of findings to the adult human spine.

Another crucial factor in terms of musculoskeletal health includes access to physical activity. In rodent studies, the amount of physical activity allowed is dependent on the housing unit. During the space shuttle era, animals were housed in NASA's Animal Enclosure Module (AEM), which provided wire grid surfaces that allowed animals to hold on to and move around their enclosures [14]. This was substantially different from Bion-M1, a 30-day mission where mice were housed in smooth-walled cylinders that limited ambulation [15]. Most recently, the AEM was adapted into the Rodent Research Habitat (RHH), which has been used in all Rodent Research (RR) missions launched on Space-X Dragon flights to the ISS [13,15]. Like the AEM, the RRH also includes internal wire grid surfaces and room for animal group housing, but unlike the AEM, the RRH has cameras to monitor animal behavior [15]. Artificial gravity has also been investigated using the Mouse Habitat Unit (MHU), an adaptation to the Japanese Aerospace Exploration Agency's (JAXA) Centrifuge-equipped Biology Experiment Facility (CBEF) that allows for centrifugation of six single-housed mice [16,17]. While this mimics gravitational loading, the MHU also utilizes smooth polycarbonate walls and wire mesh floors with minimal room for rodent ambulation [15,18]. Importantly, the AEM, Bion-M1 habitat, and RRH all support social housing, while the MHU does not.

Physical activity is particularly important for the quadruped musculoskeletal system, as loading is site specific and dependent on activity levels. Loads are highest on

the lumbar spine during physical activity, as the primary contribution to axial loading in the lumbar spine is from activation of trunk muscles [19]. In contrast, the primary contributions to axial loading in the long bones, such as the tibia, arise from both direct body weight contributions and muscle activation.

An important factor to consider is that all spaceflight experiments with rodents, except for the 5 mice that traveled to the Moon on Apollo 17, have been conducted in LEO where deep space radiation is mostly absent. Therefore, physiological effects are primarily attributed to mechanical unloading in microgravity, although uncertainty remains about potential low-level LEO radiation contributions. As future rodent experiments move to deep space beyond LEO, it is likely that elevated radiation levels will become an important factor affecting spine health, and it will be necessary to compare LEO experimental results with deep space results to determine relative effects of microgravity and deep space radiation.

Given the evolution of experimental variables used in rodent spaceflight research, a comprehensive investigation of the murine lumbar spine responses under one set of experimental conditions is warranted. To that end, the objective of this study was to characterize the bulk properties of the murine lumbar spine under Rodent Research (RR) spaceflight experimental conditions, relative to basal, ground habitat, and vivarium control conditions. To do so, lumbar spines were collected from mice after 28-days of spaceflight on the RR-10 mission to measure vertebral body and intervertebral disc parameters. Additionally, the proximal tibia was assessed to evaluate differences between a classic weight-bearing site and the load- but not weight-bearing nature of the lumbar spine. Interestingly, few differences were observed in the lumbar spine, despite large and significant differences in the proximal tibia. Our findings suggest a site specificity of tissue loss in microgravity, present in rodent models that is not observed in humans, and a potential role of physical activity in mitigating the previously observed effects of spaceflight.

## Methods

### Animals

Skeletally mature, approximately fifteen-week-old female mice (B1629SF2/J, n = 10) were flown to the ISS on SpX-21 as part of the 28-day RR-10 mission (2020). Mice were housed in the RRH modules under 12-hour light/dark cycles and provided with nutrient upgraded rodent food, water lixits, and cotton nesting material. The RRH is designed with internal wire grid walls and floors that allow the mice to push and pull themselves around all the surfaces of the enclosure [15]. Prior to the start of the experiment, mice were acclimatized to habitats conditions, including raised wire grid floors, plus flight lixits and foodbars for four weeks. Seven days and two days prior to euthanasia, mice were intraperitoneally injected with calcein to label new bone formation. At the end of the experiment, mice were euthanized on orbit with lethal injections of ketamine and xylazine. Hindlimbs were removed and the right hindlimb was immediately preserved in 4% methanol free formaldehyde fixative for 48 hours. The remaining carcasses were flash frozen and stored at -80°C until return to Earth.

Synchronous basal (n = 10), ground (n = 10), and vivarium (n = 12) control groups were also conducted at the Roskamp Institute flight support facility in Florida near the Kennedy Space Center launch site. Basal and ground control mice received four weeks of acclimation to the RRH modules prior to start of the experiment, but basal control mice were euthanized at the end of these four weeks (Experiment Day 0) to serve as a baseline control. Ground control mice were housed under identical conditions to the experimental flight mice. Basal and ground control mice were also euthanized with ketamine/xylazine. Vivarium control mice were housed under standard husbandry conditions and were euthanized by carbon dioxide inhalation and secondary cervical dislocation. Immediately after euthanasia, all control mice were dissected and preserved similarly to flight mice, with right hindlimbs preserved in formaldehyde and the remaining carcasses flash frozen for later distribution of tissues.

All animal procedures for spaceflight animals were approved by the NASA Flight Institutional Animal Care and Use Committee, and for ground control animals by the Roskamp Institute IACUC, and followed the U.S. National Institutes of Health Guide for the Care and Use of Laboratory Animals.

## Tissue collection

As part of a tissue sharing program, carcasses were thawed, and the lumbar column (L1-L5) was collected. L1-L2 discs were dissected and used for biochemical analysis. L3-L5 lumbar columns were scanned via µCT, and L4 vertebrae were used for bone microarchitectural analysis. The remaining lumbar column was used for mechanical analysis and imaging, and L3-L4 bone-disc-bone segments were tested in axial tension-compression. L5 vertebrae were further preserved and used for confocal microscopy. The preserved right hindlimb was kept intact and scanned whole, from which the proximal tibia region of interest was selected for analysis. Because freeze-thaw history has been noted to affect tissue mechanics [20], samples were minimized to three freeze thaw cycles and remained consistent across all experimental groups.

## Micro-computed tomography (µCT)

Scanning was conducted at 4.0 µm voxel size (60 kV, 166 µA) with a 0.25 mm aluminum filter using a Bruker Skyscan 1272 microCT, (Kontich, Belgium). During imaging, all samples were wrapped in saline-soaked gauze to prevent sample dehydration. Scans were reconstructed using nRecon (Bruker, vers. 1.7.4.2) with beam hardening, ring artifact, and smoothing corrections. Reconstructed scans were aligned in DataViewer (Bruker, vers. 1.5.6) and segmented and analyzed in CT-Analyser (CTAn, Bruker, vers. 1.18.8.0).

In the proximal tibia, CTAn custom processing tools were used to automatically segment trabecular bone from cortical bone. A 0.5 mm volume of interest below the epiphyseal plate was selected for analysis (n = 10–12/group). Manual quality control was conducted to confirm exclusion of cortical bone. Due to large blood vessel channels in the cortical bone of the lumbar vertebrae, automatic segmentation could not be used. Instead, vertebral body trabecular bone was manually segmented from cortical bone by drawing regions of interest (ROIs) every 20 µm and using CTAn's ROI interpolation tool. Similarly, a 0.5 mm volume of interest above the caudal growth plate was selected for analysis. For both anatomical locations, trabecular bone 3D microstructure analysis was conducted, with parameters of interest including bone volume fraction (bone volume/tissue volume, BV/TV), trabecular thickness (Tb.Th), trabecular separation (Tb.Sp), and trabecular number (Tb.N). Cortical bone 2D analysis was also conducted, with parameters of interest including total cross sectional area (Tt.Ar), cortical bone area (Ct.Ar), cortical area fraction (Ct.Ar/Tt.Ar), cortical thickness (Ct.Th), marrow area (Ma.Ar), and endocortical perimeter (Ec.Pm).

## Disc biochemistry

L1-L2 discs were used for biochemical analysis (n = 8–10/group). Two discs from each group were preserved for a separate analysis, not reported here. The remaining discs (n = 8–10/group) were carefully cut from adjacent vertebrae and lyophilized overnight (Labconco, 7740021, Kansas City, MO), after which dried samples were digested in papain (2 µg/mL). Sample digests were assayed for DNA content using the PicoGreen Assay (Invitrogen, Waltham, MA) and sulfated GAG content using the 1,9-dimethylmethylene blue assay. Aliquots of sample digests were hydrolyzed in 12N HCl overnight and assayed for OHP content using the chloramine-T spectrophotometric methods. OHP content served as an indirect measure of collagen content, as it has been previously reported that collagen and OHP content are linearly proportional [21]. GAG and OHP content were normalized to DNA content, as the small size of the discs were unreliably measured.

## Mechanical testing

L3-L4 bone-disc-bone motion segments were prepared for axial compression-tension testing by cutting through L2-L3 and L4-L5 discs, preserving the superior and inferior vertebrae. Surrounding soft tissue and musculature were cleaned off using tweezers under a dissecting microscope. Posterior and transverse processes were removed with microdissection scissors and a scalpel. Samples that were damaged during this process were not tested, resulting in reduced sample sizes (n = 7–10/group). Successfully prepared samples were potted in custom 3D printed fixtures using polymethyl

methacrylate and were lowered into place using the test instrument itself (Fig 1). Samples were hydrated in a saline-polyethylene glycol blend for 15 minutes prior to testing to maintain physiological hydration and remained submerged during testing [22].

Samples were tested under dynamic loading conditions for 20 cycles between ± 0.5 MPa at a rate of 0.1 mm/second (Instron 5943, Norwood, MA). Load limits were determined from disc geometry data collected from µCT and data from the 20th cycle was used to calculate disc mechanics. To determine the neutral zone, loading and unloading curves were averaged and the double sigmoid method was applied [23]. NZ displacement was calculated as the displacement over the neutral zone, and axial ROM was calculated as the displacement over the entire loading and unloading curve. NZ stiffness was calculated using a linear fit on the neutral zone of the averaged loading-unloading curve. Compressive and tensile stiffnesses were calculated using a linear fit to the last 20% of their maximum respective loads (Fig 2).

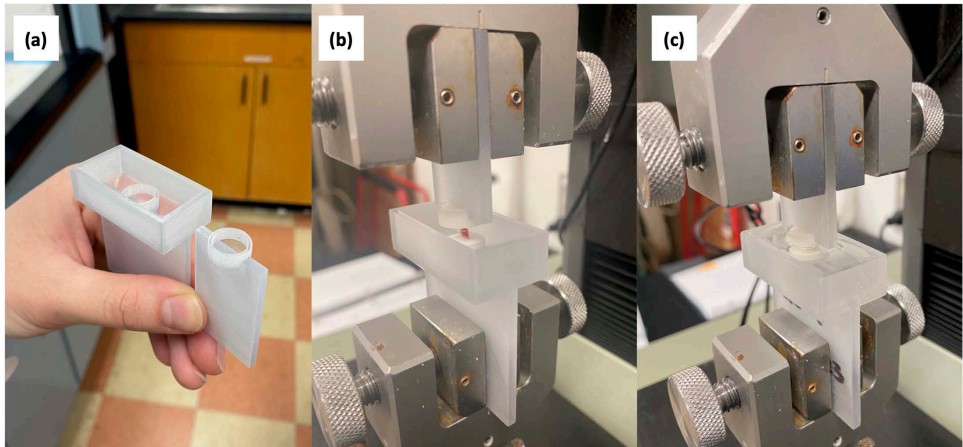

**Fig 1. Mechanical test fixturing.** a) Custom, single-use, 3-D printed fixtures were used for tension-compression testing. b) Samples were potted in place with polymethyl methacrylate (PMMA). The mechanical test instrument was used to secure and align the segment during potting. c) Once the PMMA cured, samples were submerged and hydrated for 15 minutes in saline-polyethylene glycol blend prior to testing.

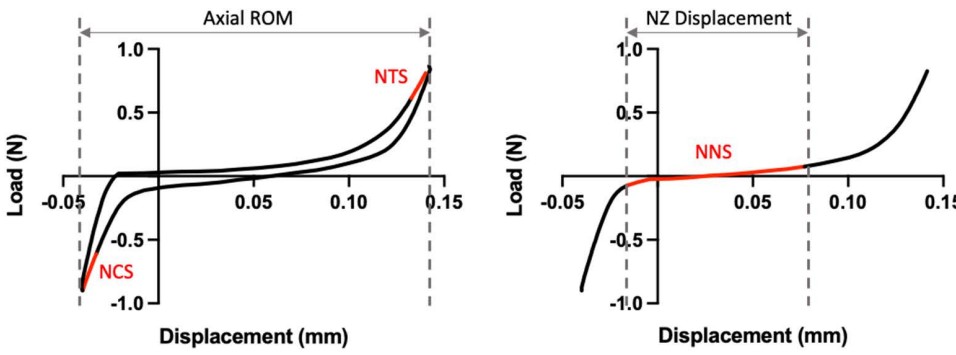

**Fig 2. Tension-compression testing curve representative of L3-L4 disc joints.** Normalized compressive stiffness (NCS) and normalized tensile stiffness (NTS) are determine from the last 20% of the respective loading curves. The neutral zone (NZ) is determined by the double sigmoid method, from which normalized neutral zone stiffness (NNS) and NZ displacement are calculated. Axial range of motion (ROM) describes the displacement over the entire loading and unloading curves.

## Calcein visualization

Once scanned and thawed, L5 vertebrae were fixed for 48 hours in 4% formaldehyde at 4°C. Following fixation, samples were washed in saline and micro-dissected along the midfrontal plane, after which marrow cells were partially brushed away to expose the bone surface for imaging and staining. After micro-dissection, vertebrae were partially dehydrated to 75% ethanol and then stained with Alizarin Red S (2 mg/ml) to label total bone mineral surface and Hoechst 33342 (100 µg/mL) to stain cell nuclei. Staining was performed for 48 hours at room temperature with gentle rocking.

After staining and washing, vertebrae were fully dehydrated in a graded series of one hour ethanol washes, followed by 24 hours in 100% ethanol. Tissue clearing was conducted over a period of one week using methyl salicylate. Vertebrae were placed on glass dishes with methyl salicylate and imaged with a Nikon FV3000 confocal microscope (UPSLAPO 20X objective; XY-axes pixel resolution 0.311 µm, Z-axis 2µm). Hoechst 33342 was imaged with UV 405nm excitation and blue 461nm emission, Calcein was imaged with blue 488nm excitation and 520nm green emission and Alizarin Red S was imaged with green 561nm excitation and red 626nm emission. Each view shown was acquired as a Z-axis stack covering about 50 µm and collapsed into a maximum intensity projection for a 2-dimensional view of the Z-stack volume of interest.

## Statistics

To determine the effect of spaceflight on biochemical, mechanical, and structural parameters, a one-way permutation ANOVA test was used. This test uses the same statistic as a one-way ANOVA, but does not assume a Gaussian distribution and is therefore appropriate for data that is not normally distributed. Briefly, the F-statistic was calculated for the original data. Then, groups were randomly resampled, and the F-statistic was recalculated; this was repeated 1000 times. P-values were calculated as the number of F-statistics greater than or equal to the experimental test statistic, plus one, divided by 1001; statistical significance was determined at $p \leq 0.05$.

When one-way permutation ANOVA indicated a significant condition effect, *post hoc* pairwise testing was conducted using permutation t-tests. This test utilizes the same approach as described above, with the t-statistic calculated instead of the F-statistic. It also offers the same advantage; it does not rely on parametric assumptions about the data. Pairwise testing was conducted between spaceflight and ground control to assess the effects of spaceflight, ground control and vivarium control to assess the effects of housing, and basal control and ground control to assess the effects of growth. Multiplicity was accounted for using the Holm-Bonferroni adjustment, with first and second thresholds of $p \leq 0.0167$ and $p \leq 0.025$, respectively.

## Results

### Lumbar vertebral bone microarchitecture

For the four trabecular parameters measured, including bone volume fraction (BV/TV, $p = 0.08$; Fig 3A), trabecular thickness (Tb.Th, $p = 0.18$; Fig 3B), trabecular separation (Tb.Sp, $p = 0.15$; Fig 3C), and trabecular number (Tb.N, $p = 0.07$; Fig 3D), condition effects were not statistically significant. Between spaceflight and ground control groups, differences were also small in magnitude; spaceflight had -11% BV/TV, -5% Tb.Th, +3% Tb.Sp, and -6% Tb.N when compared to ground control groups.

For cortical parameters studied, condition effects were significant. This includes total cross sectional area (Tt.Ar, $p < 0.001$; Fig 4A), cortical area fraction (Ct.Ar/Tt.Ar, $p = 0.004$), cortical thickness (Ct.Th, $p = 0.025$; Fig 4B), marrow area (Ma.Ar, $p < 0.001$; Fig 4C), and endocortical perimeter (Ec.Pm, $p = 0.008$; Fig 4D). At the same time, *post hoc* pairwise comparisons of cortical parameters between spaceflight and ground control were not statistically significant and small in magnitude. Comparing spaceflight to ground control groups, spaceflight samples exhibited +1% Tt.Ar, -11% Ct.Ar, -12% Ct.Ar/Tt.Ar, -12% Ct.Th, +3% Ma.Ar, and <1% change in Ec.Pm.

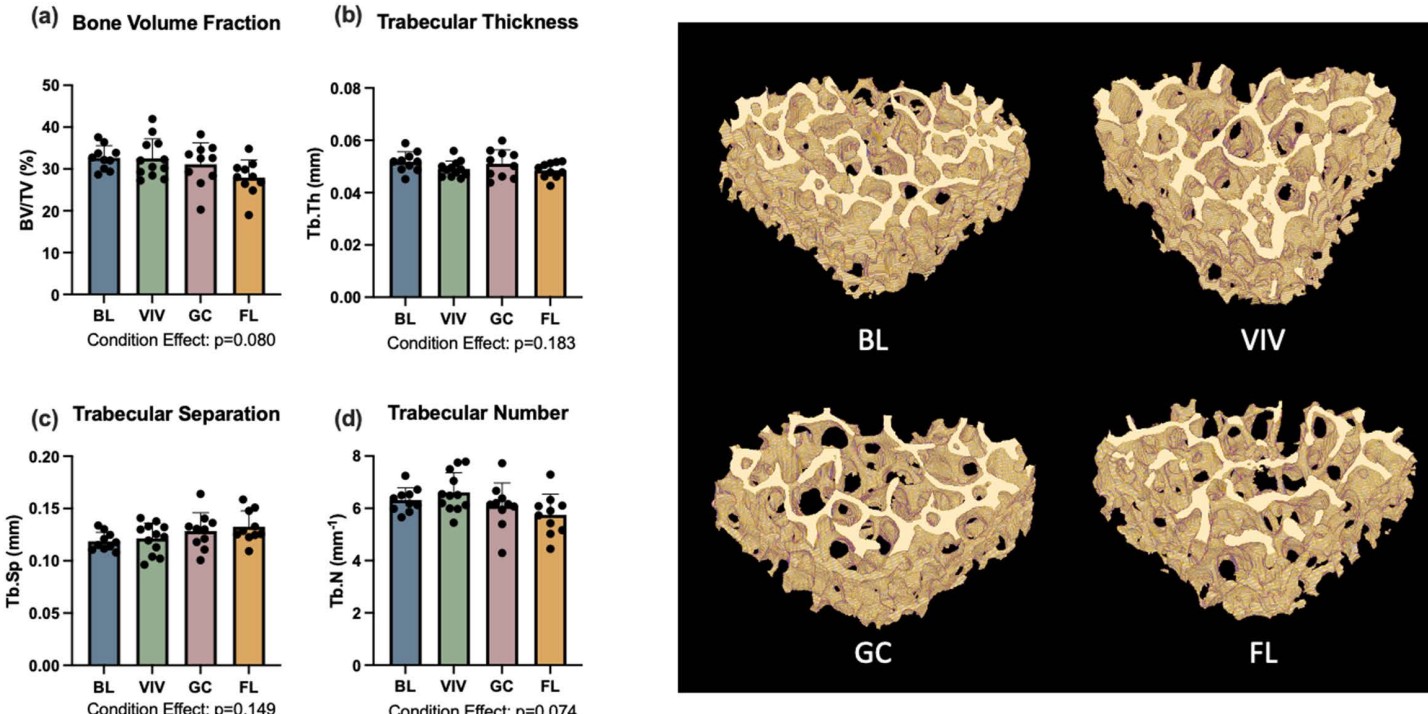

**Fig 3. L4 trabecular bone was not affected by spaceflight.** BL = Basal control, VIV = Vivarium control, GC = Ground control, FL = Spaceflight. Data is presented as mean ± standard deviation.

## Proximal tibia bone microarchitecture

The trabecular and cortical bone of the proximal tibia were both significantly affected by spaceflight. There was a 38% reduction in trabecular bone volume fraction with spaceflight (one-way ANOVA p = 0.047); however, this comparison does not reach significance after a Holm-Bonferroni correction for multiple comparisons (one-way ANOVA p = 0.024 but no significant differences in *post hoc* analysis; Fig 5A). Trabecular thickness was significantly affected by growth (-9%, p = 0.026), housing (-19%, p < 0.001), and spaceflight, with trabecular thickness nearly 20% lower in the spaceflight group than the ground control (p < 0.001; Fig 5B). The one-way ANOVA results for trabecular separation also demonstrated significant condition effects (p = 0.034; Fig 5C), but the *post hoc* analysis did not reach significance for differences between groups. Condition effects were also significant for tibia cortical area (p = 0.001; Fig 6B) and cortical area fraction (p = 0.007; Fig 6C), with spaceflight tibias showing 21% less cortical area (p = 0.003) and 18% less cortical area fraction (p = 0.003) than ground control samples.

## Disc biochemistry

L1-L2 discs were analyzed for sulfated glycosaminoglycan (GAG), oxidized hydroxyproline (OHP), and DNA content, where OHP is an indirect measure of collagen content. GAG and OHP content were normalized to DNA content, as the wet weight of the small discs could not be precisely measured. OHP to GAG ratio was also calculated, as previous studies have suggested changes in OHP to GAG ratio after spaceflight [7,8]. The effect of experimental condition was not statistically significant for any of the parameters measured, including DNA, GAG/DNA, OHP/DNA, and OHP/GAG ratios. Spaceflight samples had an average of 1771.8 ± 294.2 ng of DNA (Fig 7A), 21.94 ± 2.79 µg GAG/µg DNA (Fig 7B), and

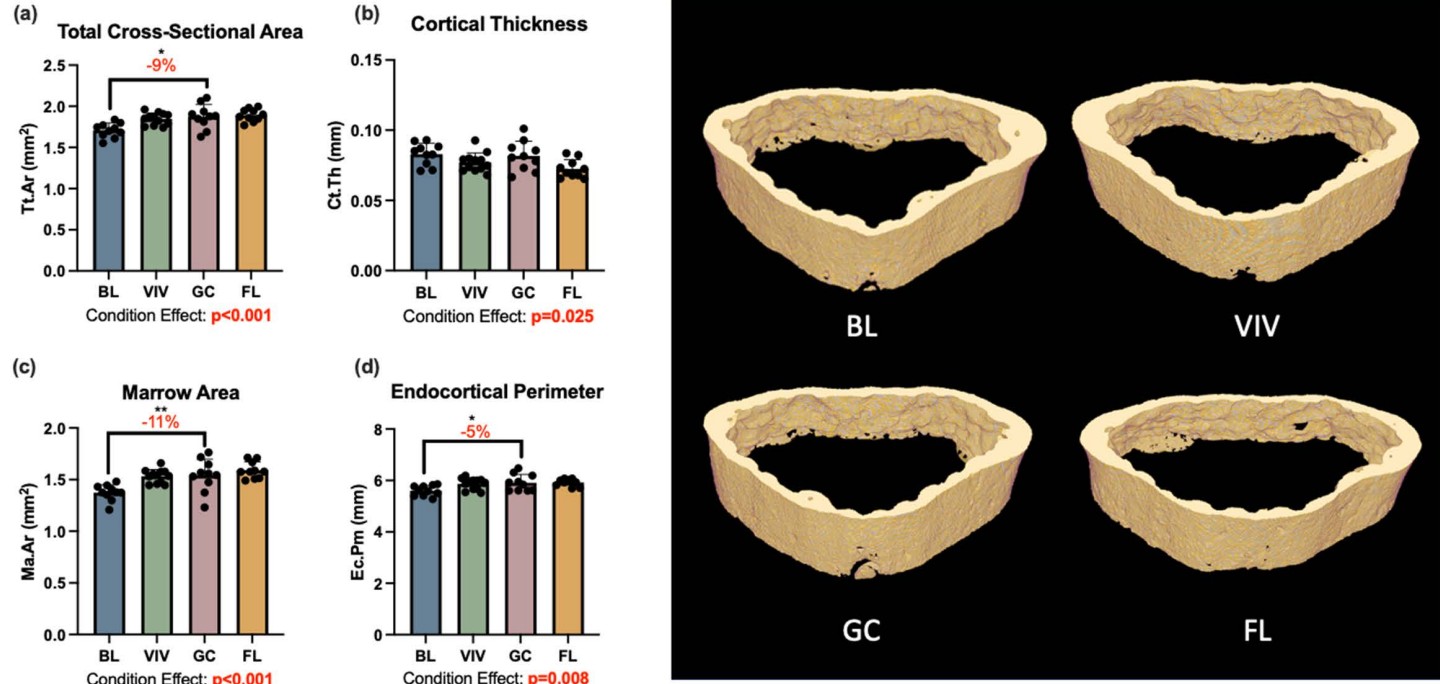

**Fig 4. L4 cortical bone was significantly affected by growth over the experimental period, but not by spaceflight.** BL = Basal control, VIV = Vivarium control, GC = Ground control, FL = Spaceflight, *p < 0.05 **p < 0.01. Data is presented as mean ± standard deviation.

15.59 ± 2.31 µg OHP/µg DNA (p > 0.4) (Fig 7C). The average OHP/GAG ratio for spaceflight samples was 0.71. The differences between spaceflight and ground control averages for these four parameters did not exceed 5%.

## Disc joint biomechanics

L3-L4 disc joint mechanics were assessed under tension-compression dynamic loading and data was analyzed for normalized compressive, tensile, and neutral zone stiffnesses, as well as neutral zone displacement and total axial range of motion (Fig 1). One-way ANOVA results noted a statistically significant difference in normalize compressive stiffness (NCS) between groups (p = 0.049); however, pairwise comparisons for growth, housing, and spaceflight did not reach statistical significance (Fig 8A). Notably, the average NCS for spaceflight samples was 15% smaller than that of the ground control samples.

Condition effects for the remaining four mechanical parameters did not reach statistical significance and *post hoc* pairwise statistical tests were not conducted. However, differences in magnitude were evident. Spaceflight samples had an average normalized tensile stiffness (NTS) of 3.64 ± 0.53 MPa, which was 14% smaller than that of ground control samples (Fig 8B). Similarly, spaceflight samples had an average normalized neutral zone stiffness (NNS) of 0.22 ± 0.06 MPa, which was 35% smaller than that of ground control samples (Fig 8C). In both displacement parameters, neutral zone (NZ) displacement and axial range of motion (ROM), differences were much smaller. Spaceflight samples had an average NZ displacement of 0.057 ± 0.009 mm, which was only 4% larger than that of ground control samples (Fig 8D). In axial ROM, spaceflight samples exhibited more compliance, with an average axial ROM (0.18 ± 0.01 mm) that was 12% larger than that of ground control samples (Fig 8E).

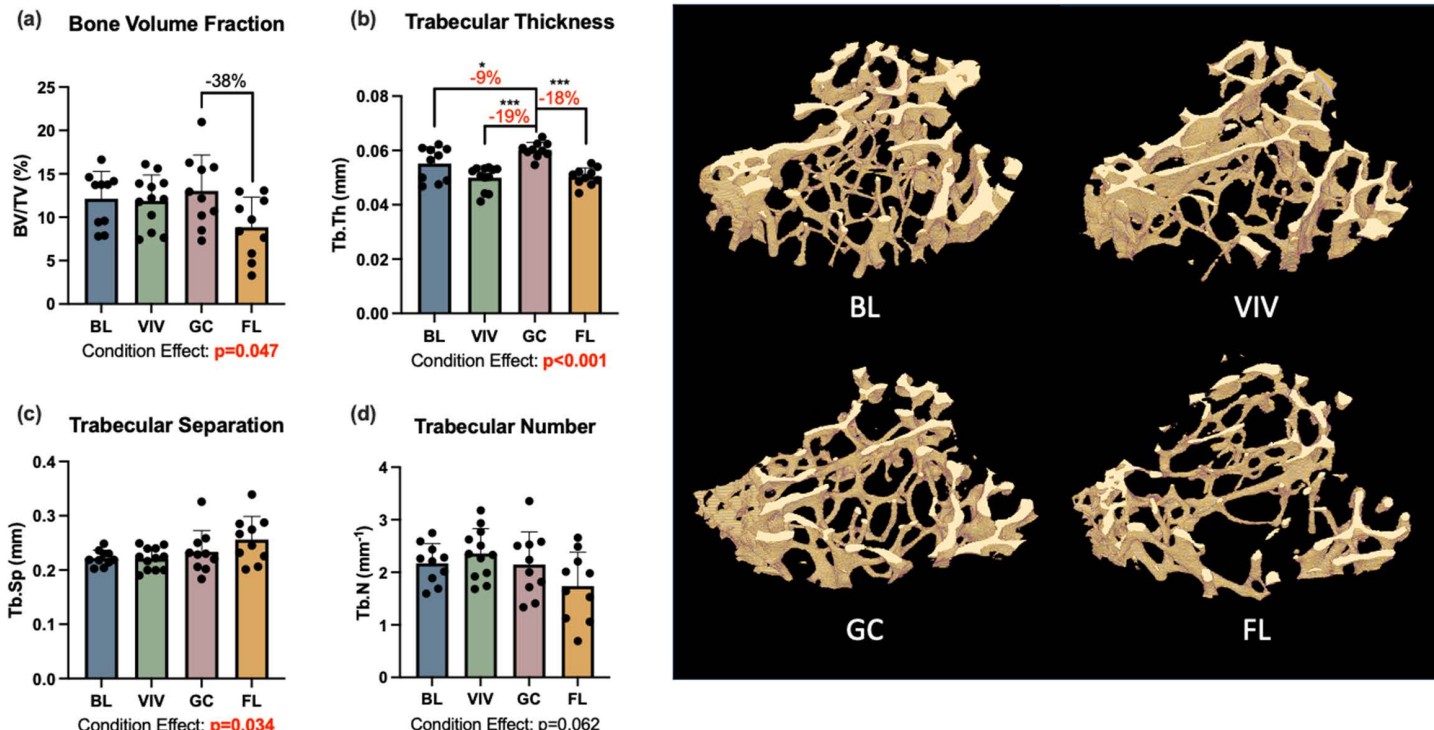

**Fig 5. Proximal tibia trabecular bone was significantly affected by spaceflight.** BL = Basal control, VIV = Vivarium control, GC = Ground control, FL = Spaceflight, *p < 0.05, **p < 0.01, *** p < 0.001. Data is presented as mean ± standard deviation.

## Discussion

Rodent spaceflight research offers a unique window into the relationship between mechanical unloading and spine health, but the prior literature is composed of a highly diverse set of experiments, animal models, and flight missions that make the drawing of general conclusions challenging. As rodent studies continue to provide insight into space biology for applications to human flight, confidence in the appropriate animal model is necessary. Furthermore, evaluating the utility of the rodent spaceflight model for the spine requires robust characterization of bulk properties of the spine and comparison to another, well characterized, anatomical location. Thus, the aim of this study was to comprehensively assess the effects of spaceflight in LEO on the vertebral bone microarchitecture, intervertebral disc biochemistry, and intervertebral disc joint biomechanics of the murine lumbar spine under current standard spaceflight experimental conditions.

In this study, bone loss in the L4 vertebrae with spaceflight was minimal and statistically insignificant. This aligns with recent studies of murine vertebral bone after Rodent Research-4 (RR-4), another 30-day spaceflight, which reported significant experimental conditions that did not reach statistical significance in *post hoc* pairwise conditions [5,6]. The maintenance of bone microarchitecture in the lumbar spine is in contrast with changes in bone of the proximal tibia. In the proximal tibia, trabecular thickness was reduced in a large and statistically significant manner between spaceflight and ground controls (-18%, p < 0.001). Bone volume fraction was also largely reduced in spaceflight samples (-38% vs. ground control), though this was statistically not significant at the first significance threshold determined by the Holm-Bonferroni correction for multiplicity (p = 0.024). These findings also align with those from RR-4, where reductions in trabecular bone volume fraction (-32%), trabecular thickness (-14%), and trabecular number (-14%) were observed at statistically significant or near statistically-significant levels [5].

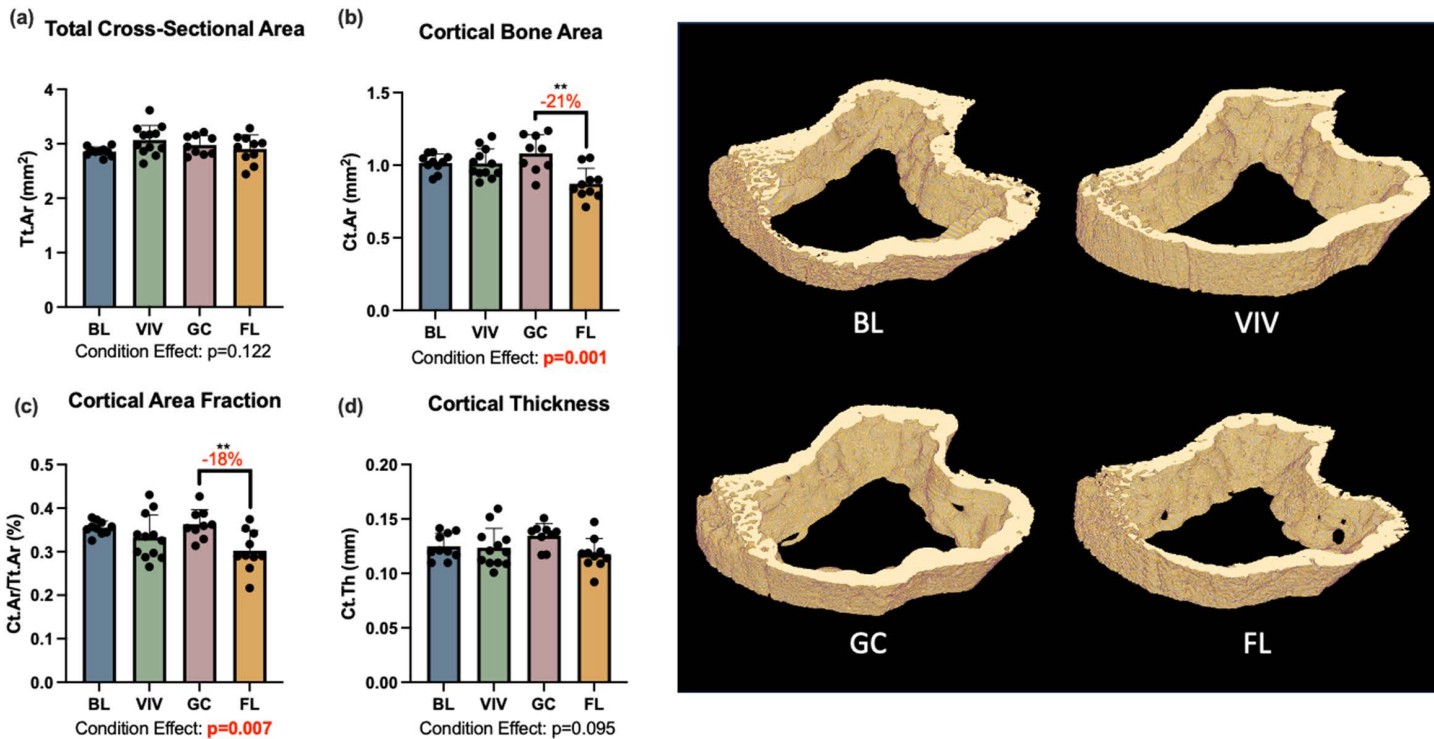

**Fig 6. Proximal tibia cortical bone was significantly affected by spaceflight.** BL = Basal control, VIV = Vivarium control, GC = Ground control, FL = Spaceflight, **p < 0.01. Data is presented as mean ± standard deviation.

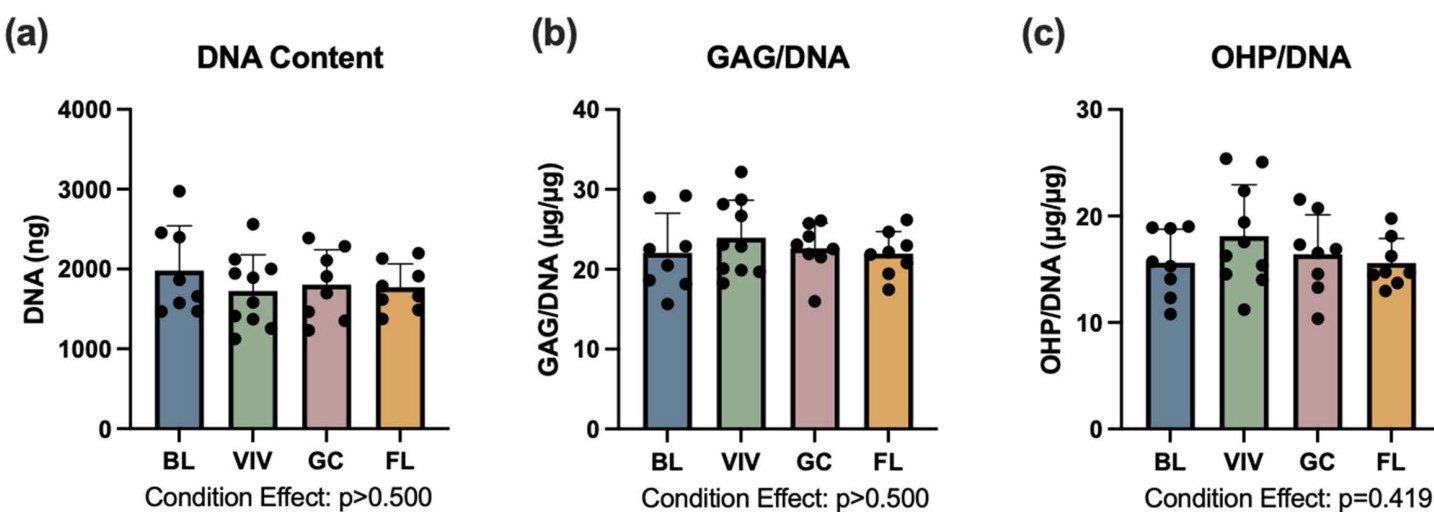

**Fig 7. L1-L2 disc biochemistry is not significantly affected by experimental condition.** BL = Basal control, VIV = Vivarium control, GC = Ground control, FL = Spaceflight. Data is presented as mean ± standard deviation.

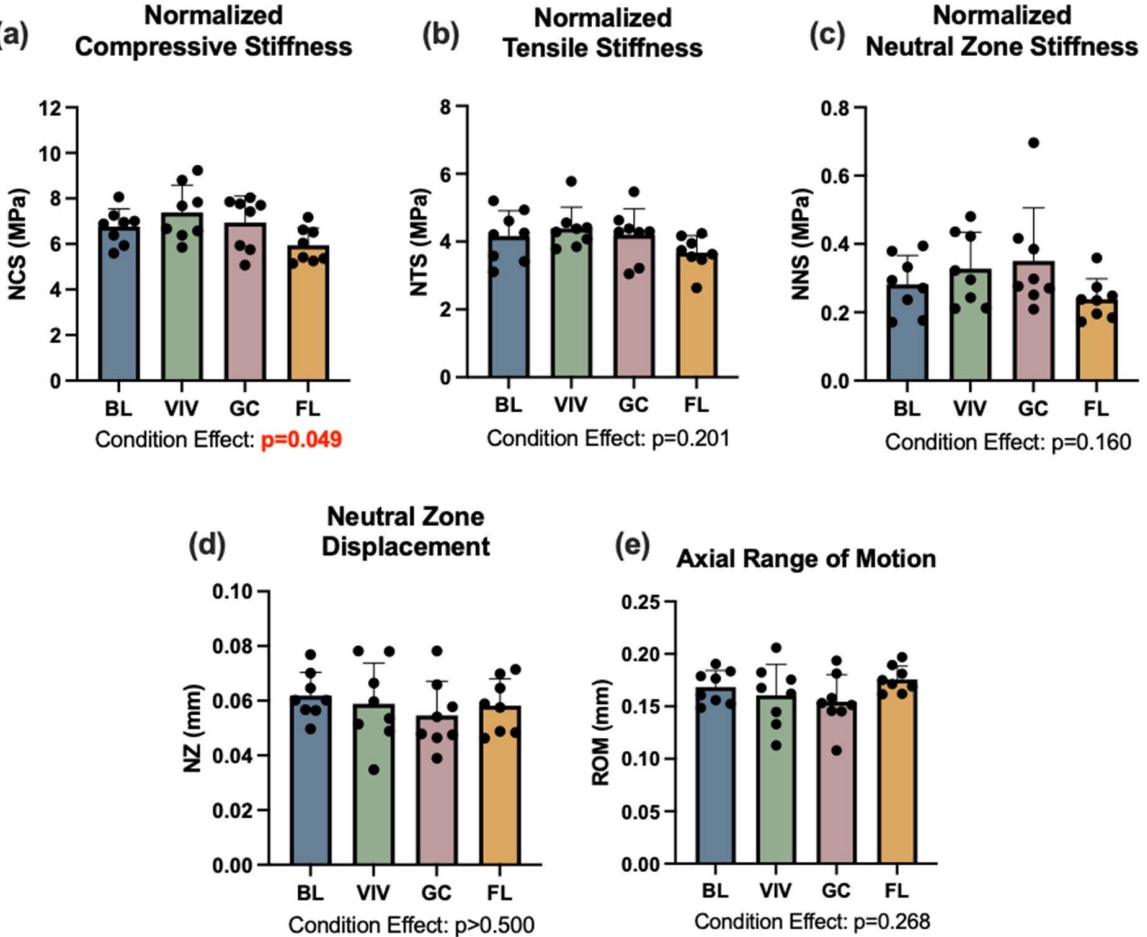

**Fig 8. Disc joint mechanics were not affected by spaceflight.** An experimental condition effect observed for normalized compressive stiffness (NCS) did not result in significant pairwise differences when *post hoc* testing was conducted. BL=Basal control, VIV=Vivarium control, GC=Ground control, FL=Spaceflight. Data is presented as mean±standard deviation.

The site-specific nature of spaceflight induced bone loss has been increasingly revealed, with recent studies demonstrating substantial bone loss in weight-bearing bones, like the femur and the tibia, and maintenance of bone tissue in non-weight bearing sites, such as the calvarium and the mandible [5,12]. While the lumbar vertebrae are typically considered as weight-bearing for the purpose of such studies, data presented here may indicate that site specificity extends to the lumbar spine, with a distinction between load-bearing and weight-bearing sites. While both the tibia and lumbar spine are load-bearing, the majority of loading on the lumbar spine in the quadruped animal results from activation of trunk muscles, rather than directly from the weight of the animal [19]. Thus, in the case of unloading via the removal of gravitational loading and body weight, unloading may disproportionally affect a weight-bearing site over one that is load- but not predominantly weight-bearing.

It is important to recognize that spaceflight induced changes in lumbar vertebral bone have been reported. Gerbaix et al. reported large and significant losses (~35–55%, p<0.01) in L3 vertebral bodies after Bion-M1, a 30-day spaceflight mission [4]. However, an important characteristic of the Bion-M1 experiment was the rodent housing hardware, consisting of smooth-walled cylinders with no surface features that would allow mice to grab on to for ambulation [14]. On the

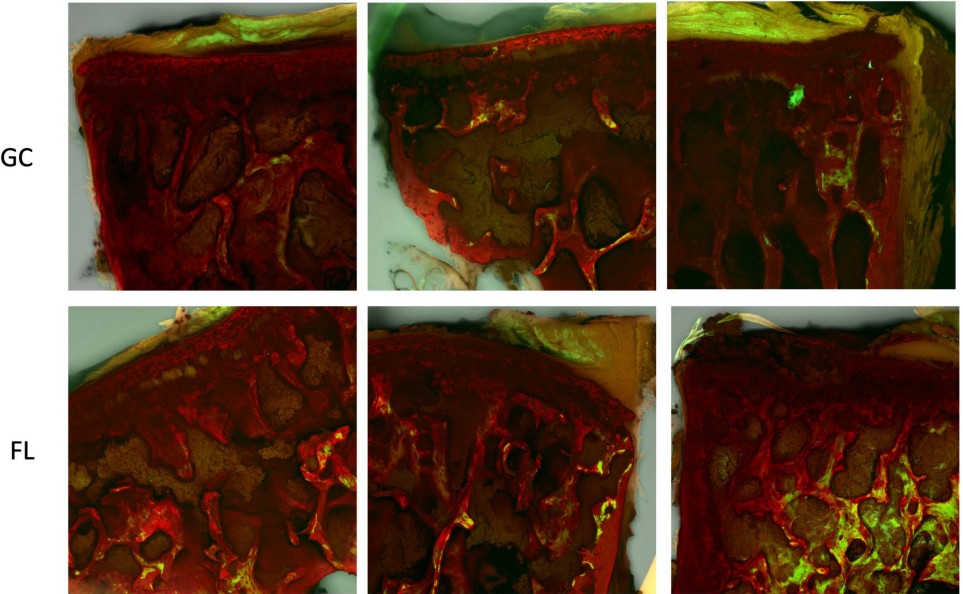

GC

FL

**Fig 9. Bone mineralized in the final week of the experiment is visualized with calcein (green) and dual stained with Alizarin Red S.** While not quantified, images indicate some spaceflight samples exhibit new bone formation (white arrow, green) despite microgravity conditions. Each image is a different sample; GC = Ground control, FL = Spaceflight.

contrary, RR animals are provided with wire mesh that can be easily handled by mice and used to navigate around the enclosure [15]. Furthermore, recent characterization of rodent behavior in the RR habitats has demonstrated a shift from forelimb ambulation to quadruped ambulation as the rodents acclimate to microgravity, as well as the development of a high-speed running in circles behavior that continues for the duration of the experiment [13]. This running behavior was also observed towards the end of this experiment and can be viewed in S1 Fig. Since axial loading of the murine spine is primarily dependent on muscle activity rather than direct bodyweight, we hypothesize that the increased physical activity in the RH may have loaded the lumbar spine. Theoretically, increased mechanical loading from high-speed circle running and other ambulation could protect the lumbar spine from spaceflight induced unloading, support the maintenance of bone homeostasis, and potentially encourage formation of new bone.

To better understand the balance between bone resorption and formation during the last quarter of the mission, a brief preliminary assessment was conducted to assess *do novo* bone formation in the L5 vertebra. As part of the primary RR-10 investigation, animals were injected with calcein 7 days and 2 days prior to euthanasia. Confocal images suggest that some spaceflight animals experienced bone growth during the final week of the flight (Fig 9), further supporting the hypothesis that increased physical activity in the final week of the flight may mitigate microgravity induced unloading in the lumbar spine. However, additional investigations into this hypothesis are necessary, as the current study did not examine time-dependent changes nor quantify new bone formation.

Changes in intervertebral disc biochemistry are supportive of the hypothesis that unloading of the lumbar spine was either incomplete or ameliorated by increased activity levels. Changes in disc biochemical content were neither material nor statistically significant, with differences between spaceflight and ground control groups never exceeding 5%. Had unloading of the lumbar spine been completely achieved, decreases in proteoglycan content would be expected, as ground based unloading studies in rats have demonstrated reduced proteoglycan content after 3 or more weeks of tail suspension [24,25]. Additionally, proteoglycan synthesis has been demonstrated after 7 days intermittent of dynamic

loading in the mouse disc, suggesting that increased activity in the second half of the experiment could be enough to encourage proteoglycan synthesis [26]. Thus, the minimal changes observed here may indicate incomplete unloading of the lumbar spine, or recovery of the biochemical composition with increased loading.

The biomechanics of the disc joint with spaceflight are less clear. Statistically, spaceflight did not have a significant effect on any of the five mechanical parameters studied. This aligns with the finding that spaceflight did not affect disc biochemistry, as biochemical composition and mechanical function of the disc are interconnected. However, in percent difference alone, spaceflight discs were less stiff in tension, compression, and the neutral zone. This would be expected with decreased proteoglycan content, as increased proteoglycan content has been linked with increasing neutral zone stiffness [27]. One could also speculate that while biochemical composition could recover during the second half of the experiment, the macroscale joint mechanical properties would not return to baseline as quickly. Importantly, these differences did not reach statistical significance, this may simply be due to small sample sizes and limited statistical power.

It is worth noting that the strain of mice used in this study (B6129SF2/J) is a second filial generation from a C57BL/6J and 129S1/SvImJ hybrid strain, selected to control for a genetic knockout also studied by the primary investigators of the RR-10 mission. While bone homeostasis patterns differ slightly between strain [28], there is limited knowledge on the relationship between mechanical loading, bone growth, and mouse strain. Additionally, mice were 14.5 weeks at the beginning of the experiment, just reaching skeletal maturity. This is evident in differences between basal and ground control vertebral cortical bone; total cross-sectional area, marrow area, and endocortical perimeter were significantly larger in ground control animals compared to basal animals, indicating endosteal resorption [29]. However, because cortical thickness and cortical area fraction were similar between basal and ground control groups, results suggest bone formation at the periosteal surface in combination with assumed endosteal resorption [29]. This is typical for mice of this age (4–5 months) and represents normal skeletal remodeling [28,30]. Nonetheless, mouse strain and age may be additional differentiating factors between results presented here and previous observations.

As with most spaceflight studies and biospecimen sharing collaborations, this study is limited both by sample size and sample transfer logistics. Small sample sizes in this study (n = 8–12) may have limited the statistical power, potentially resulting in Type II error. Specifically, the small sample size may have contributed to results where the one-way ANOVA test suggested statistical differences between groups, but the *post hoc* analysis did not determine any differences between individual groups (Prox. Tib Trabecular BV/TV, Fig 5A; NCS, Fig 8A). As noted above, findings from this work suggest that the housing and animal activity may contribute to preserving spine health; however, we were not able to control for either factor. This is another limitation of the current study design.

Regardless of these limitations, the findings presented here are of importance to our understanding of the rodent spine in spaceflight. They extend the current knowledge of microgravity-based unloading site specificity to the lumbar spine and highlight the difference between load- and weight-bearing anatomical locations. They also suggest that physical activity may be an important factor in considering how spaceflight affects the lumbar spine, particularly due to contributions of internal muscle activation to axial loading of the lumbar spine. Lastly, it illustrates the potential for a relationship between recovery and disc health with increasing physical activity. In all, it opens the door for further investigations to differentiate between the effects of mechanical unloading and serve as a reference to study other spaceflight stimuli such as deep space radiation.

## Supporting information

**S1 Fig. Video of mice throughout spaceflight.**
(MP4)

**S1 Table. Summary of microCT, mechanical, and biochemical data.**
(XLSX)

## Acknowledgments

The authors would like to acknowledge Rachel Kui for her support with preliminary analysis of the lumbar vertebrae. We would also like to acknowledge the NASA Ames Research Center Bone Lab and the RR-10 Principal Investigator Biospecimen Sharing Program Team (C. Juran, E. Blaber, R. Globus, Y. Shirazi-Fard, and E. Almeida) for providing spine and tibia samples for this study..

## Author contributions

**Conceptualization:** Shiyin Lim, Eduardo A.C. Almeida, Grace D. O'Connell.

**Data curation:** Shiyin Lim, Joanna E. Veres.

**Formal analysis:** Shiyin Lim, Joanna E. Veres, Eduardo A.C. Almeida, Grace D. O'Connell.

**Funding acquisition:** Eduardo A.C. Almeida, Grace D. O'Connell.

**Investigation:** Shiyin Lim, Joanna E. Veres.

**Methodology:** Shiyin Lim, Joanna E. Veres, Eduardo A.C. Almeida, Grace D. O'Connell.

**Project administration:** Eduardo A.C. Almeida, Grace D. O'Connell.

**Resources:** Eduardo A.C. Almeida, Grace D. O'Connell.

**Supervision:** Eduardo A.C. Almeida, Grace D. O'Connell.

**Visualization:** Shiyin Lim, Eduardo A.C. Almeida, Grace D. O'Connell.

**Writing – original draft:** Shiyin Lim, Eduardo A.C. Almeida, Grace D. O'Connell.

**Writing – review & editing:** Shiyin Lim, Eduardo A.C. Almeida, Grace D. O'Connell.

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
