## [Decision Letter · Decision Letter 0]

23 Oct 2024

PONE-D-24-10223Characterization of the murine spine for spaceflight studiesPLOS ONE

Dear Dr. O'Connell,

Thank you for submitting your manuscript to PLOS ONE. After careful consideration, we feel that it has merit but does not fully meet PLOS ONE’s publication criteria as it currently stands. Therefore, we invite you to submit a revised version of the manuscript that addresses the points raised during the review process. The revision process highlighted that there are issues concerning two specific aspects, i.e., sample size (power analysis required) and improvements to be done for Introduction and Discussion sections, which currently lack to be specific enough.

We look forward to receiving your revised manuscript.

Kind regards,

Alessandra Aldieri

Academic Editor

PLOS ONE

“The authors would like to acknowledge Rachel Kui for her support with preliminary analysis of the lumbar vertebrae. We would also like to acknowledge the NASA Ames Research Center Bone Lab and the RR-10 Principal Investigator Biospecimen Sharing Program Team (C. Juran, E. Blaber, R. Globus, Y. Shirazi-Fard, and E. Almeida) for providing spine and tibia samples for this study. This work was funded by the National Science Foundation (1751212, GDO) and NASA (NNH14ZTT001N 14-14SF Step2-0063 to E. Almeida). The funders played no role in study design, data collection, analysis and interpretation of data, or writing of this manuscript.”

“This study was funded by the National Science Foundation (1751212, GDO) and the National Aeronautics and Space Administration (NNH14ZTT001N 14-14SF Step2-0063, EACA). The funders played no role in the study design, data collection, analysis and interpretation of data, decision to publish, or preparation of this manuscript.”

Reviewers' comments:

Reviewer's Responses to Questions

**Comments to the Author**

1. Is the manuscript technically sound, and do the data support the conclusions?

Reviewer #1: Yes

Reviewer #2: Yes

2. Has the statistical analysis been performed appropriately and rigorously? 

Reviewer #1: No

Reviewer #2: Yes

3. Have the authors made all data underlying the findings in their manuscript fully available?

Reviewer #1: Yes

Reviewer #2: Yes

4. Is the manuscript presented in an intelligible fashion and written in standard English?

Reviewer #1: Yes

Reviewer #2: Yes

5. Review Comments to the Author

Reviewer #1: The authors explored the possible changes of the mechanical, structural and biochemical properties due to spaceflight in rodents’ spine and tibia.

The study is really interesting, well organised and relevant.

I have two concerns:

1. Sample size: the authors clearly explained the possible problem related with the sample size. However, the obtained evidence resulted different from the ones available in the literature. Could the authors perform a power analysis in order to understand if the sample size is sufficient for their aim?

2. Title, introduction and discussion are generic and do not help the readers in understanding the state of the art and the knowledge provided by this study. The introduction is generic and does not help in focalising the attention to the topic of the paper. The authors explained the confusion generated by the different flight durations, different houses, different physical activities, the complexity in understanding the results and translating them for the human research. However, they did not perform a factorial design to take into account all these parameters. They complained about the large amount of data already available and partially useful but, in the current form, their work seems another data contribution. I suggest confining the arguments of the discussion in the effects of house and flight. The same for the title which does not help in understanding the paper topic. Finally, the discussion could be improved providing a more clear and extensive explanation of the obtained results.

Minor comments:

Lines 46-47: specify that the “previous studies” were performed on humans.

Lines 103-107: why here this sort of conclusion?

Line 123: could you specific the type and age of the mice in the control groups?

Line 154: could you specific the approach used for the bone segmentation? Is it a single level threshold?

Lines 185-186: why were posterior and transverse processes removed?

Line 190: Why were the specimens tested submerged? Can this option increase the hydration of the IVD?

Lines 228-235: Did the authors check the gaussian distribution? Why did they perform a PERMANOVA instead of a Kruskal-Wallis?

Line 255: if data did not meet the gaussian distribution, median and interquartile range should replace the mean and standard deviation

Line 270 and following: why did you refer to the one-way ANOVA if another test was performed?

Line 305: I suppose Fig. 2

Line 378: de novo

Lines 410-412: did you perform a power analysis?

Reviewer #2: This is an interesting study that looks at the effects of unloading on the balance between resorption and formation and the subsequent influences that it has on weight/load-bearing bones.

What was the age of the rodents in the Bion-M1, a 30-day spaceflight mission? Could that have played a role in the difference in lumbar spine response?

6. PLOS authors have the option to publish the peer review history of their article (what does this mean?). If published, this will include your full peer review and any attached files.

Reviewer #1: No

Reviewer #2: No

---

## [Author Response · Author response to Decision Letter 1]

3 Jan 2025

We thank the reviewers for their thoughtful comments, which have improved the manuscript in both rigor and clarity. The comments have been incorporated into the revised manuscript in blue and are noted in the responses below.

Editorial Comments

1. Please ensure that your manuscript meets PLOS ONE’s style requirements, including those for file naming.

We thank the editors for the reminder to adhere to the style requirements. We have reviewed the requirements, double-checked the submission, and have updated the manuscript and files accordingly. Files are now named and referenced appropriately, in-text citations use brackets instead of parentheses, funding information has been removed from the acknowledgements, and supporting information has been added to the end of the manuscript.

2. Plesae remove any funding-related text from the manuscript and let us know how you would like to update your Funding Statement.

Funding information has been removed from the acknowledgements section. We do not need to revise the funding statement but thank the editors for the opportunity to do so.

3. We note that you have indicated that there are restrictions to data sharing for this study. For studies involving human research participant data or other sensitive data, we encourage authors to share de-identified or anonymized data. However, when data cannot be publicly shared for ethical reasons, we allow authors to make their data sets available upon request.

a. If there are ethical or legal restrictions on sharing a de-identified data set, please explain them in detail (e.g., data contain potentially identifying or sensitive patient information, data are owned by a third-party organization, etc.) and who has imposed them (e.g., a Research Ethics Committee or Institutional Review Board, etc.). Please also provide contact information for a data access committee, ethics committee, or other institutional body to which data requests may be sent.

There are no ethical or legal restrictions on sharing de-identified data, and we have updated our data availability statement accordingly.

b. If there are no restrictions, please upload the minimal anonymized data set necessary to replicate your study findings to a stable, public repository and provide us with the relevant URLs, DOIs, or accession numbers. Please see http://www.bmj.com/content/340/bmj.c181.long for guidelines on how to de-identify and prepare clinical data for publication. For a list of recommended repositories, please see https://journals.plos.org/plosone/s/recommended-repositories. You also have the option of uploading the data as Supporting Information files, but we would recommend depositing data directly to a data repository if possible. Please update your Data Availability statement in the submission form accordingly.

We have opted to upload our data as Supporting Information files. The data has been including as Supporting Information S2, labeled S2 File 2 at the end of the manuscript and uploaded as S2_File 2.xlsx.

4. Please include captions for your Supporting Information files at the end of your manuscript, and update any in-text citations to match accordingly.

We thank the editors for the reminder to include this information, and have added the section “Supporting Information” at the end of the manuscript, with captions for supporting information.

Reviewer 1 Comment

Sample size: the authors clearly explained the possible problem related with the sample size. However, the obtained evidence resulted different from the ones available in the literature. Could the authors perform a power analysis in order to understand if the sample size is sufficient for their aim?

We thank the reviewer for suggestion, which correctly highlights a limitation of the study. After consultation with our statistics collaborator (Assistant Professor of Statistics, Amanda Glazer at UT Austin), we refrained from conducting a post hoc power analysis because statistics research has shown that post hoc power analyses are logically flawed and can be misleading when used as an aid to interpret non-significant results. We added text in the acknowledgement section to reference our consultation with Professor Glazer (lines 444-445). We added language describing our hesitation with post hoc power analysis on lines 422-424. This is highlighted in the following references:

Zhang Y, Hedo R, Rivera A, Rull R, Richardson S, Tu XM. Post hoc power analysis: Is it an informative and meaningful analysis? Gen Psychiatry. 2019;32(4):3–6.

Heckman MG, Davis JM, Crowson CS. Post Hoc Power Calculations: An Inappropriate Method for Interpreting the Findings of a Research Study. J Rheumatol. 2022;49(8):867–70.

Title, introduction and discussion are generic and do not help the readers in understanding the state of the art and the knowledge provided by this study. The introduction is generic and does not help in focalising the attention to the topic of the paper. The authors explained the confusion generated by the different flight durations, different houses, different physical activities, the complexity in understanding the results and translating them for the human research. However, they did not perform a factorial design to take into account all these parameters. They complained about the large amount of data already available and partially useful but, in the current form, their work seems another data contribution. I suggest confining the arguments of the discussion in the effects of house and flight. The same for the title which does not help in understanding the paper topic. Finally, the discussion could be improved providing a more clear and extensive explanation of the obtained results.

We agree with the reviewer that the original title was vague. The revised manuscript includes a more descriptive title that highlights the main takeaway from this research.

The intention behind describing previous studies was to highlight the myriad of experimental conditions used to study rodent spine biomechanics with spaceflight. These variations prevent researchers from combining and cross-referencing results from different studies, thus impeding development of a comprehensive understanding of the effects of spaceflight on the spine. However, the reviewer is correct in pointing out that this results in the introduction being less focused. Therefore, we have removed this text from the Introduction (originally lines 57-94) and focused on the studies already described in the discussion.

Minor comments:

Lines 46-47: Specify that the “previous studies” were performed on humans.

We appreciate the suggestion to improve clarity and have added “astronaut” to indicate that these studies were performed on humans.

Lines 103-107: Why here this sort of conclusion?

We appreciate the suggestion and understand why this conclusion may seem out of place. The text has been modified clarify conclusions from this work (lines 87-91).

Line 123: Could you specific the type and age of the mice in the control groups?

We thank the reviewer for the opportunity to improve clarity. The control mice were the same strain as flight mice and were age-matched. This information is now listed on line 109.

Line 154: Could you specific the approach used for the bone segmentation? Is it a single level threshold?

Bone segmentation was conducted via adaptive thresholding in CTAn.

Lines 185-186: Why were posterior and transverse processes removed?

Poster and transverse processes were removed to ensure all load was transferred through the intervertebral disc, as opposed to through facet joints. This allows us to normalize load to the area of the disc, allowing for testing load limits to be adjusted to the size of intervertebral disc. Additionally, removal of the posterior and transverse processes is standard practice for testing of intervertebral disc mechanics. We added a few citations to this effect on line 171.

Line 190: Why were the specimens tested submerged? Can this option increase the hydration of the IVD?

Specimens were submerged to preserve hydration of the intervertebral disc, as disc mechanical properties are highlight dependent on tissue hydration. To prevent hyperphysiological swelling, we used a saline-polyethylene glycol (S-PEG) blend over phosphate buffered saline, which has been shown to over-hydrate the disc (supporting citation listed in text, line 177. Preliminary work was conducted to determine the best ratio of saline to PEG to prevent hyperphysiological swelling. The methods text was modified to clarify the purpose of the S-PEG blend (lines 177-178).

Lines 228-235: Did the authors check the gaussian distribution? Why did they perform a PERMANOVA instead of a Kruskal-Wallis?

We thank the reviewer for their question. We did check for Gaussian distribution, and the datasets did not satisfy a normal distribution assumption, likely due to the relatively small sample size. We performed a permutation ANOVA over a Kruskal-Wallis to provide a more robust measure of statistical significance over non-parametric ranked tests. The effect of permutation based statistical analysis is also highlighted in our previous work with our statistical collaborator, Professor Philip Stark in Statistics at UC Berkeley:

Glazer AK, Luo H, Devgon S, Yao X, Siwei S, McQuarrie F, Li Z, Palma A, Wan Q, Gu W, Sen A, Wang Z, O’Connell GD, Stark PB. Look Who’s Talking: Gender Bias in Academic Job Talks. ScienceOpen 03/2024.

We did not perform a PERMANOVA test, which is used to understand multivariate data by calculating Euclidean distances. In this case, we performed a permutation ANOVA because we aimed to understand the effects of one independent variable (experimental condition) on each outcome measure. If we had an additional independent variable (such as mouse genotype), we agree with the reviewer that the PERMANOVA would have been a preferred approach.

The methods text was modified for additional clarity in our statistical approach (lines 20-221).

Line 255: If data did not meet the gaussian distribution, median and interquartile range should replace the mean and standard deviation

We thank the reviewer for pointing out this important statistical distinction. We now present median (IQR) throughout the manuscript and on the figures. The percent differences between groups have also been recalculated, using the median instead of the mean. The changes in percent differences were minimal and do not change the conclusions of the paper. Changes are reflected in the manuscript in blue.

Line 270 and following: Why did you refer to the one-way ANOVA if another test was performed?

We thank the reviewer for pointing out this inconsistency. We adjusted the text to state “permutation test” in place of one-way ANOVA. These changes can be seen on lines 260, 261, 265, 296, and 425 in blue.

Line 305: I suppose Fig. 2

We see that the language originally used, “respective loading curves”, is a bit vague and can be confusing. We changed the figure caption to specifically reference compressive and tensile loading curves to improve clarity. These changes can now be seen on 195.

Line 378: de novo

This typo has been corrected.

Lines 410-412: did you perform a power analysis?

A power analysis was not performed a priori based on the nature of this study. This study was performed as part of a tissue sharing program with NASA researchers, where limitations in physical space, cost, and animal support limit the number of animals that can be used for each study.

Reviewer 2 Comments

This is an interesting study that looks at the effects of unloading on the balance between resorption and formation and the subsequent influences that it has on weight/load-bearing bones.

We thank the reviewer for taking the time to review our manuscript.

What was the age of the rodents in the Bion-M1, a 30-day spaceflight mission? Could that have played a role in the difference in lumbar spine response?

Mice in the Bion-M1 mission were 19-20 weeks at launch, or 4-5 months old. This is slightly older than the mice used in this study, which were 14.5 weeks at launch. While the 4–6-week age difference is notable, both cohorts of mice were considered skeletally mature. As such, we do not believe age to have been the primary driver of the differences between studies. Nonetheless, we have added a sentence to clarify this in the discussion (line 355-356).

---

## [Decision Letter · Decision Letter 1]

10 Mar 2025

28 days in space reveals minimal effects of microgravity on murine lumbar vertebrae and intervertebral discs

PONE-D-24-10223R1

Dear Dr. O'Connell,

We’re pleased to inform you that your manuscript has been judged scientifically suitable for publication and will be formally accepted for publication once it meets all outstanding technical requirements.

Kind regards,

Alessandra Aldieri

Academic Editor

PLOS ONE

Additional Editor Comments (optional):

Reviewers' comments:

Reviewer's Responses to Questions

**Comments to the Author**

1. If the authors have adequately addressed your comments raised in a previous round of review and you feel that this manuscript is now acceptable for publication, you may indicate that here to bypass the “Comments to the Author” section, enter your conflict of interest statement in the “Confidential to Editor” section, and submit your "Accept" recommendation.

Reviewer #1: All comments have been addressed

Reviewer #3: (No Response)

2. Is the manuscript technically sound, and do the data support the conclusions?

Reviewer #1: Yes

Reviewer #3: Yes

3. Has the statistical analysis been performed appropriately and rigorously? 

Reviewer #1: Yes

Reviewer #3: Yes

4. Have the authors made all data underlying the findings in their manuscript fully available?

Reviewer #1: Yes

Reviewer #3: (No Response)

5. Is the manuscript presented in an intelligible fashion and written in standard English?

Reviewer #1: Yes

Reviewer #3: Yes

6. Review Comments to the Author

Reviewer #1: All comments have been addressed by the authors.

Just two minor comments:

- double check the use of RHH and RRH on the manuscript,

- data reported in the Supporting Information should be reported with an appropriate number of significant figures.

Reviewer #3: I read with interest the revised version of the manuscript entitled " 28 days in space reveals minimal effects of microgravity on murine lumbar vertebrae and intervertebral discs".

The authors have appropriately addressed all the criticalities identified in the first version of the manuscript. The revised version provides additional details, which facilitate the understanding of the conducted work.

I consider the revised version of the manuscript acceptable for publication

Minor comments

Line 70-75: The “Rodent Research Habitat” is sometimes shortened in RRH and sometimes in RHH

7. PLOS authors have the option to publish the peer review history of their article (what does this mean?). If published, this will include your full peer review and any attached files.

Reviewer #1: No

Reviewer #3: No

---

## [Editor Report · Acceptance letter]

PONE-D-24-10223R1

PLOS ONE

Dear Dr. O’Connell,

I'm pleased to inform you that your manuscript has been deemed suitable for publication in PLOS ONE. Congratulations! Your manuscript is now being handed over to our production team.

Kind regards,

on behalf of

Dr. Alessandra Aldieri

Academic Editor

PLOS ONE